# Exploring the Evolution of Human Resource Analytics: A Bibliometric Study

**DOI:** 10.3390/bs13030244

**Published:** 2023-03-10

**Authors:** Eithel F. Bonilla-Chaves, Pedro R. Palos-Sánchez

**Affiliations:** 1School of Business Administration, Technological Institute of Costa Rica, Cartago 30109, Costa Rica; 2Department of Financial Economy and Operation Management, Faculty of Economics and Business Sciences, University of Sevilla, 41018 Sevilla, Spain

**Keywords:** strategic HRM practices, HR analytics, people analytics, bibliometric analysis, management and organisational behaviour, bibliometrix

## Abstract

The objective of this study is to identify and analyze the most relevant scientific work being undertaken in HR analytics. Additionally, it is to understand the evolution of the conceptual, intellectual, and social structure of this topic in a way that allows the expansion of empirical and conceptual knowledge. Bibliometric analysis was performed using Bibliometrix and Biblioshiny software packages on academic articles indexed on the Scopus and Web of Science (WoS) databases. Search criteria were applied, initially resulting in a total of 331 articles in the period 2008–2022. Finally, after applying exclusion criteria, a total of 218 articles of interest were obtained. The results of this research present the relevant notable topics in HR analytics, providing a quantitative analysis that gives an overview of HR analytics featuring tables, graphs, and maps, as well as identifying the main performance indicators for the production of articles and their citations. The scientific literature on HR analytics is a novel, adaptive area that provides the option to transform traditional HR practices. Through the use of technology, HR analytics can improve HR strategies and organisational performance, as well as people’s experiences.

## 1. Introduction

In the very competitive environment of the corporate world, it is increasingly important that human resource management (HRM) is performed effectively to achieve corporate success; in this context, strategic HRM (SHRM) is the implementation model employed to manage human resources (HR) along with the activities aimed at allowing the company to achieve its objectives [1].

This area covers all the major decisions about HR practices, the composition of the group of human capital resources, the specification of required behaviours, and the measurement of the effectiveness of the decisions derived from the various business strategies and/or competitive situations encountered [1]. The composition of the group of human capital resources is a collective phenomenon and human creation that is based on organizations and information, so organizations transmit information [2].

This reasoning allows us to propose HR analytics as a novel system to collect, analyze, and present this information from organizations. Using the compendium of definitions made by [3], They propose that HR analytics is an information- and technology-enabled HR practice that uses descriptive, visual, and statistical analyses of data related to HR processes, human capital, organisational performance, and external economic benchmarks to establish business impact and to enable data-driven decision making.

A variety of terms are used in this subject matter, such as “Workforce analytics”, “Talent analytics”, “People analytics”, “Human Capital analytics”, “Human Resource analytics”, and “HR analytics”. The authors of [3] indicate in this respect that the most frequently used term in the literature is HR analytics, although this should still be considered to be an emerging term. Likewise, “People analytics” has been identified as another term of much interest that is used frequently; the set of terms mentioned above has therefore been included in the scope and analysis of this study. The work conducted by [4] defines People analytics as an area of HRM practice, research, and innovation related to the use of information technologies and descriptive and predictive data analysis that employs visualization tools to generate useful information about the dynamics of the workforce, human capital, and individual and team performance that can be used strategically to optimise the effectiveness, efficiency, and results of an organisation, as well as to improve the experience of employees.

The following research questions from this study are presented in Table 1 below:

This new study seeks to give rise to and suggest new ideas for continued increasing research on this subject matter, in the hope of providing a guide as to the practical application of the adoption and use of HR analytics for evidence-based decision-making at the organisational and individual level, at the same time as supporting the increasingly strategic alignment of HR operations [5].

To answer these questions, this article has the main objective of identifying and analyzing the scientific literature in the area of HR analytics. Additionally, it seeks to understand the evolution of the conceptual, intellectual, and social structure of this subject in a way that allows the expansion of empirical and conceptual knowledge.

A literature review was therefore carried out by means of bibliometric analysis, consulting the scientific production on HR analytics academic articles indexed in the Web of Science and Scopus databases and analyzing the articles and emerging trends in research published between 2008 and 2022.

This article is organised as follows: Section 1 presents the research topic to be investigated, along with the study’s purpose, objectives, and research questions. Section 2 includes the literature review for the bibliometric analysis. Section 3 explains the scientific methodology used, by means of the Science Mapping Workflow and the Bibliometrix software. This is followed by the analysis of the results and the later discussion of these. Finally, the conclusions are presented, and possible future lines of research are suggested.

## 2. Literature Review

There exists great and increasing interest in the literature on HR analytics. Exploring the orientation and dynamics of the gradual transformation of this subject is therefore worth conducting by means of reviewing the current state-of-the-art in HR analytics.

Among the studies undertaken to review this development in the academic theory and research on the subject, the research performed by [6] suggests that HR professionals should pay attention to four key points in HR analytics: (a) HR professionals need to develop a strategic understanding of how people contribute to the success of their organisation; (b) Analytics should be based on a deep understanding of data and the context in which it is collected in order to generate meaningful insight. This allows the generation of significant metrics, which in turn enable the measurement and modelling of the costs and benefits of different HR strategies and methods; (c) These metrics and tools should allow the identification of the key talent segments, those groups of employees whose performance makes the most strategic difference to the business and its performance; (d) Data-based decision-making should be derived after careful empirical analysis is made using advanced statistical and econometric techniques that go beyond the analysis of the correlation between variables used in experiments, such that identification is made of the way that human capital contributes to the organisation’s performance.

The authors of [3] further explain that People analytics is a term that has arisen from Google, which uses it to describe its data-driven approach to HRM. Google’s success has popularised the concept as a best practice in HRM, given that it is used by the world’s leading companies to improve their competitive advantage as mentioned by [5]. It is for this reason that Google’s Project Oxygen has been a success story since 2010, as explained by [6] and referenced by [7] as a good example of incorporating data analytics into day-to-day decision-making, in a way that has helped to obtain crucial knowledge about people operations. Therefore, we can say that HR analytics enjoys great popularity [7,8]. However, some studies warn of the risks of HR analytics [9].

It is in this context that [7] refers at once to both the concepts of People analytics and HR analytics as the use of analytical techniques such as data mining, predictive analytics, and contextual analysis to enable managers to make better workforce-related decisions. Nonetheless, the HR analytics literature remains in a state of constant transformation. The authors of [8] explain that the use of bibliometric analysis allows an understanding of the evolution of the state-of-the-art of a specific area in the existing literature to be able to discover emerging trends through the performance of articles and journals, patterns of collaboration, research components, and the exploration of intellectual structure. Previous bibliometric analyses of HR analytics by [9,10] conclude that this domain is in an incipient or emerging stage.

Table 2 presents previous reviews related to the topic of this study. As can be seen, this research is focused on articles, early access, and reviews and extends the databases consulted to Web of Science with the 2022 year included.

The authors of [19] recommend six steps for organisations to take into consideration in promoting HR analytics: (a) The development of an analytics strategy in a way that takes into account current and future needs; (b) The identification of key questions or investment decisions on which to focus; (c) Focussing these questions on future-oriented issues, not past ones; (d) Not settling on the use of the data at hand; (e) Performing data cleansing; (f) Limiting challenges to data validity by means of standardised data definitions and processes in the generation of reports and analyses.

On the other hand, [20] has elaborated and provided the following five moderating factors for HR analytics: (a) Problem identification: HR professionals must be able to identify organisational problems and ask the right questions; (b) Data infrastructure: HR analytics requires that data that area accessible, accurate and consistent across functions, even including those external data to the organisation; (c) Information technology: This must be appropriate to advanced analysis and focus on data exploration, analysis, and modelling to effectively perform HR analytics; (d) Analytical skills: HR analytics requires professionals with specific skills to prepare the data, perform statistical analysis, and communicate the results in a meaningful and understandable way; (e) Business focus: To implement HR analytics effectively, the business focus must be comprehensive, integrating processes, data, and analytics throughout the organisation.

Despite the progress and efforts made in studying HR analytics, [21,22] reiterate that there remains a shortage of rigorous quantitative and qualitative empirical studies on the results of HR analytics or People analytics. Nonetheless, this study identifies indications that some quantitative empirical studies in HR analytics are beginning to emerge.

## 3. Materials and Methods

The potential to combine the best available academic evidence with the judgement and experience of practitioners in the true tradition of evidence-based practice can be obtained through the methodology of systematic review [14]. According to [23], recognising trends in the analyses of thematic areas is possible by using bibliometrics as an indicator, which can reveal the development of trends in basic structures.

Thus, for this study, bibliometric analysis [24] was carried out using the general Science Mapping Workflow methodology described by [25], as shown in Figure 1. The application and organisation of the bibliometric analysis were carried out by means of the standard workflow consisting of five steps [26].

In the data collection stage, information was obtained from the Web of Science (WoS) and Scopus databases. This was performed using a Scientific Mapping Workflow for bibliometric analysis over a 14-year period, between 2008 and 2022. This was performed to complete the systematic review of the literature proposed by [27], in which the search strategy filters the relevant criteria using the PRISMA methodology [28]. This methodology details the phases of identification in the databases, the selection of records, and the filtering of elements by the eligibility criteria that have been employed.

As shown in Table 3, for the databases and search criteria applied, a total of 331 academic articles related to “HR analytics” were identified after applying the PRISMA methodology [28] to find the documents pertaining to this investigation. The inclusion parameters used in the databases consisted of seven main keywords: “People analytics”, “HR analytics”, “Human Resource analytics”, “Workforce analytics”, “Talent analytics”, “Employee analytics”, and “Human Capital analytics” [3,4,29] for the period from the year 2008 to the year 2022 (July).

The keywords had to appear in the title, abstract, and the keywords themselves of the articles consulted. The search results could only include articles and research reviews. Other selection parameters were also included, such as the incorporation of a filter to include only articles in English, and those that had been published or that had gone through the editorial and/or peer-review process.

The exclusion parameters used to delimit the content of the articles and related documents excluded documents that were not research or scientific review articles. Similarly, articles in languages other than English were excluded. The selected articles had to have a clear relationship with or contribute to the field of study of HR analytics. Likewise, the main objectives and research questions of the articles had to be clearly described and explained.

Once the results of the databases were obtained, the records of each database were exported in the BibTeX plain text file format [30] to maintain consistency between data sources, to later be able to combine both files into a single file for processing. Both WoS and Scopus databases allow records to be exported directly in the standard BibTeX bibliographic format; however, each database includes the different fields in a different order.

This meant that the databases had to be standardised, starting with the records being converted into a dataframe in R-Studio [31], then concatenating the records regardless of the database they came from, removing duplicates [32]. This process eliminated 113 duplicate records from the results obtained from the databases, arriving at a final total of 218 articles. This final result of records in a single database was processed using R statistical software.

Data analysis was made by applying the scientometric methodology for the bibliometric analysis of science mapping using the Bibliometrix software [33], as other recent work in the field of human resources has been conducted [34,35]. This is supported by the Biblioshiny web interface, also developed by [33] and available from the Comprehensive R Archive Network (CRAN). The reasons for choosing this software are based on a recent work [36], which indicates that Bibliometrix contains the most comprehensive and appropriate set of techniques.

This Bibliometrix R software package must be installed and loaded by executing the “library(bibliometrix)” command in R-Studio [31]. Immediately following this, it is necessary to execute the command “biblioshiny()” and load the Biblioshiny web interface, which provides a graphic visualisation of data and statistics. For the purpose of this study, the graphic information corresponds to HR analytics according to the parameters defined.

## 4. Results

### 4.1. General Summary of the Bibliographic Collection Processed

Subsequently, the analysis and standardisation phase of the Scientific Mapping Workflow procedure was undertaken. Table 4 shows the overview of the research data. It can be highlighted that in the 14-year period that was analyzed, 218 articles were identified as a result of excluding duplicates.

These articles arose from 134 different sources, with an annual average publication rate of three articles per year and an average number of 10.4 citations. Similarly, 9390 articles, 652 keywords, and 45 different authors were referenced. This detail demonstrates how the study of HR analytics is an emerging field and how it manages to maintain or inspire interactions with other topics.

This behaviour can be observed in Figure 2, which shows that the number of scientific publications on HR analytics begins to increase from the year 2014, some years after what could be considered the starting point of its popularity [37]. This research explains the six key ways to track, analyze, and use employee data, ranging from establishing simple metrics that monitor the overall health of the organisation to identifying talent shortages and excesses long before these occur.

Consecutively over the following years, research in HR analytics showed an average annual growth rate of 1.8%, with an accentuated growth peak between 2016 and 2017. Notable among the publications of the year 2017 are the peer-reviewed article by [3] and the publication by [38], which proposes 4 clusters of analytical maturity for companies, with these companies belonging to the innovative disruptive analysis cluster. This cluster commenced using analytics earlier, applying more complex techniques and more advanced applications such as HR analytics, where its use is more common and shows a higher level of analytical maturity.

It can therefore be said that HR analytics research has shown sustained growth since 2017. In the year 2019, there is also notable growth in publications, including an article published by [39] that details the way a new generation of HR professionals is developing an“HR stack”, which includes other management frameworks to increase HR competencies, among these HR analytics.

An exception can be seen in the decline shown in 2020, at the time of the COVID-19 pandemic. There is also notable growth in publications in the year 2021, which include the publication by [40] of a literature review of 60 years of research on the relationship between technology and HRM. This explains that in the final proposed time period, from 1997 to 2019, there was increased interest in making better use of the HR data accumulated in HR information systems (HRIS) for business decision-making, with this, therefore, representing the growing field of HR analytics.

Similarly, Figure 2 shows the linear regression of variance with an explanatory effect coefficient of 82.6% for scientific publications per year, representing a positive relationship through the interpretation of [41,42], thus reflecting the validity and accuracy of the research topic.

### 4.2. Thematic Evolution

The thematic evolution of the keywords related to HR analytics and the most relevant authors on this topic a revisualised in the Sankey diagram [43] shown in Figure 3. This indicates the order of magnitude of the various information flows of the quantitative data for the main topics. The indexing of the content represents the redundant visualisation of the quantity of relationships with authors, highlighting the increased connection of the terms “HR analytics” and “Artificial Intelligence”.

It could be asserted that consolidation is made of the greater use of term “HR analytics” with respect to other related key terms used in publications on the same subject. Among the authors, it is notable that Steven McCartney together with Na Fu relate to most of the main HR analytics topics, with both of these authors having published very recent articles in HR analytics [44] and People analytics [17].

The first article addresses whether HR analytics can increase organisational performance, affirming that access to HR technology is a precursor of HR analytics. The other article provides a systematic review of the literature on People Analytics. Other authors, including Gonen Singer, Dan Avrahami, and Hila Chalutz Ben-Gal, have made use of the term “Artificial Intelligence” together with the term “Machine Learning” for application in HR analytics [45].

In the study conducted, a comprehensive framework of analysis is proposed that can serve as a support tool for the making of decisions by HR recruiters in real-world environments to improve hiring and placement processes. The prediction approach uses the machine learning model, applying the Variable-Order Bayesian Network model.

#### 4.2.1. Relevant Sources

The most relevant databases were used for the bibliometric analysis. Table 3 shows that there was a greater number of results from Scopus (193) than from WoS (138). Table 5 shows the most relevant scientific sources by the number of articles published on HR analytics. The most relevant scientific journals on the subject of HR analytics were identified in the period analyzed, with an average of two articles published.

At 10 articles each, the Human Resource Management Journal and the Journal of Organizational Effectiveness: People and Performance were the journals that published the most articles on HR analytics, followed by Human Resource Management with eight articles. The journals with the highest number of publications on these topics were journals with a focus on HR.

The 10 most cited journals for the topic of HR analytics are presented in Table 6, with Human Resource Management being the journal that tops this list with a total of 212 citations. Followed by the International Journal of Human Resource Management and the Academy of Management Journal with 188 and 154 citations, respectively. The journals with the highest number of citations on these topics were journals related to HR and Business, like Harvard Business Review. These represent the most cited journals for the topic of HR analytics.

The most important journals on the topic of HR analytics can be identified by applying Bradford’s law [46] as shown in Figure 4. These core sources are identified in zone 1, the shaded area that includes the following journals: the Human Resource Management Journal and the Journal of Organizational Effectiveness: People and Performance. These journals are at the core [47] of HR analytics and include the most relevant research on the topic, so they should be given special importance when preparing publications on this subject.

As shown in Table 7, the highest impact factor is consistent with Bradford’s law, through the journal Human Resource Management with an h-index [48] of 8 and 168 citations. This journal started publishing on the topic of HR analytics in 2018. This is followed by the Journal of Organizational Effectiveness: People and Performance, which began publishing on this topic in 2017, and which has an h-index of 7 with 144 citations. These journals have the greatest level of impact of all those publishing on HR analytics.

In terms of the increase in publications, the journal Personnel Review stands out, showing exponential growth as seen in Figure 5. This growth commenced in 2019 and remains on the rise even in the first months of 2022. Included among the HR analytics research contained in this journal are the publications of [49,50].

The article by [49], “An ROI-based review of HR analytics: Practical implementation tools” conducts a literature review of HR analytics based on ROI (return on investment) and has 22 citations. This article provides the practical application of this quantitative measurement tool for managerial decision-making, as motivated by the limited high-quality research in the field. At the same time, this ROI-based perspective can provide increased opportunities for the practical adoption of HR analytics.

In addition, of note, the article by [50], “The ethics of people analytics: Risks, opportunities and recommendations” has 10 citations. This article performs a “scoping review” of HR analytics to understand the ethical considerations and recommendations to be taken into account for ethical practice in this matter. These recommendations are (a) Transparency and equity; (b) Legal compliance; (c) Ethical guidelines and statutes; (d) Proportionality and protection; (e) Data rights and consent; (f) Inclusion of data subjects; (g) Skills and people culture; (h) Evaluation; (i) Ethical business models. In contrast, the Harvard Business Review shows a clear decrease in publications, while the Human Resource Management International Digest has begun a sudden reduction in publications.

#### 4.2.2. Relevant Authors

The authors with the most publications on the topic of HR analytics are Caryl Charlene Escolar-Jimenez from the University of Tokyo in Japan, Reggie C. Gustilo from De La Salle University in the Philippines and KichieMatsuzaki from the University of Tokyo in Japan, as shown in Table 8. These authors have published five articles, with the coincidence that for all three authors, the article with the highest number of citations is “A Neural-Fuzzy Network Approach to Employee Performance Evaluation”, published in 2019 with ten citations.

This work applied the artificial intelligence technique called “artificial neural networking” using the neuro-fuzzy profiling system to optimise traditional employee performance evaluations. This allows HR departments and decision-makers in organisations to easily identify the strengths and weaknesses of employees for professional promotion, training, and development in achievement, leadership, and behaviour, in contrast to the subjectivity of the traditional system [51]. The most popular research areas by the authors in HR analytics are computer science, data science, and organizational behaviour.

The scientific output of Hila Chalutz Ben-Gal from the Afeka Tel Aviv Academic College of Engineering in Israel as of 2019 has been continuously focused on the topic of HR analytics, as shown in Figure 6. In 2019, her first article was “An ROI-based review of HR analytics: Practical implementation tools”.

Similarly, Steven McCartney from Maynooth University in Ireland has been active in the publication of HR analytics articles since 2020. In that same year, he published the article “21st century HR: A competency model for the emerging role of HR Analysts” with five citations, in which he explores the key competencies and KSAOs (knowledge, skills, abilities, and other characteristics) required for the role played by HR Analysts [52].

The frequency of publications per author in any field of research is known as Lotka’s law [53]. Table 4 shows that of the 461 authors identified for this study, 86.3%, which are 398 authors, have a publication on HR analytics, as shown in Table 9. Following the Pareto principle, 10% of the authors wrote two articles and 2.2% contributed three articles. In contrast, there are only four and three authors who published four and five articles, respectively.

In accordance with [54], Figure 7 shows that 86.3% of the authors wrote only one article on HR analytics and that only 0.7% of the authors wrote five articles on this. It can therefore be presumed that the majority of the authors have published in the field due to the novelty of the topic.

With an h-index of four, the authors Escolar-Jimenez C., Gustilo R., and Matsuzaki K., who published the article “A Neural-Fuzzy Network Approach to Employee Performance Evaluation”, have the highest impact factor of all HR analytics authors, with this being higher than the average of two as shown in Table 10.

These are followed by Boudreau J., GuerryM., and Ben-Gal H. with an h-index of three for the first and two for the latter two authors. For Boudreau J., in addition to the article “An evidence-based review of HR Analytics” published in co-authorship, another publication in 2014 is noteworthy, this being the article “HR strategy: Optimizing risks, optimising rewards” which has 12 citations. This article suggests that in the field of HR, instead of minimising or controlling unwanted results in dealing with risks, a balanced approach to risk-taking is required for the optimisation there of [55].

Guerry M. and Ben-Gal H. are the authors who come in the middle of the ranking. Noteworthy for Guerry M. among the articles published in co-authorship is the 2018 article, “Predicting voluntary turnover through human resources database analysis”, which has 14 citations. This study determines that by using a priori only available data from reliable HR databases, valuable predictions regarding staff turnover can be generated for use by HR managers to prevent and reduce voluntary turnover more reliably [56].

The author Ben-Gal H., on the other hand, published the article “An ROI-based review of HR analytics: Practical implementation tools”. For all these authors mentioned, a total of 317 citations are added for articles published related to HR analytics.

The universities to which the authors belong are shown in Table 11. Notable among these is Bar-Ilan University in Israel, which has ten publications, followed by Tilburg University in the Netherlands and the University of Southern California in the United States of America with eight articles each. In addition, the Copenhagen Business School has six publications, while the remaining universities mentioned presented five articles each.

The scientific production by country shown in Table 12 uses the SCP indicator to show that the USA with 43 articles leads the number of publications on HR analytics by country. It is also the country that shows the highest rate of collaboration with an MCP of four. This is followed by India and the United Kingdom, with 25 and 11 articles published per country, respectively.

In similar fashion, the USA maintains the highest number of article citations by country, with 933 representing an average of 21.7% of citations, as can be seen in Table 13. It is followed by India with 223 citations, the United Kingdom with 204 citations, and the Netherlands with 164 citations. Among the countries mentioned, there are a total of 1524 article citations per country related to HR analytics publications.

#### 4.2.3. Relevant Articles

The articles with the most citations are presented in Table 14. The first is the article by [6] with 147 citations and an average yearly citation rate of 21 times. This study reveals that the development of HR analytics is hampered by the lack of understanding of the analytical thinking of HR professionals and HR analytics teams.

The article, therefore, suggests that HR professionals should pay attention to improving their skills and knowledge to become “champions” of this new approach, such that HR analytics methods can make HR transcend into having strategic influence at the managerial level in order to benefit the organisation and its employees.

The second most cited article with 122 citations is by [57]. This article explains that among the domains used to specify where HR investments should be directed, a move should be made to an external-internal approach, in which HR reacts to the challenges of the organisation to participate more fully in the development of strategy and value-adding. The authors propose that HR analytics should be created in a way that focuses on the right problems.

The article by [58] is in third place with 121 citations and an average annual citation rate of 24.2. The paper presents a case study using HR analytics, which was undertaken using the Smart HR 4.0 analysis methodology to identify employees at risk of attrition. In addition, it promotes linking the concept of Smart HR 4.0 to the digital transformation of HR functions based on a “science of people”.

With 117 citations, the article by [37] in the Harvard Business Review is the fourth most cited article. This paper reports that leading companies such as Google, Best Buy, P&G, and Sysco use sophisticated data collection and analysis technology to get the most value from their talent. It further includes six key ways to track, analyze, and use employee data.

The article by [3] providing a peer-reviewed literature review comes in fifth place with 113 citations. In sixth place with 101 citations is the paper by [59]. In the empirical research these authors conducted, development is made of a model to examine HR analytics practices along with an incentive system that produces greater productivity when the practices are implemented collectively rather than separately. Detailed data on the adoption of HR software are also included.

Finally, [60] authored the seventh article with 74 citations. This paper uses two case studies to illustrate how HR analytics can deliver value by forming an ongoing part of the management of end-to-end decision-making. Included among the suggestions made are proposals to commence with the business problem, to take HR analytics outside of HR, to remember the “human” side of HR, and to train HR professionals to have an analytical mindset.

The premises, suggestions, and orientation of these articles provide direction as to where the efforts of HR analytics should be focused to transcend beyond research into the subject matter so as to evolve into value-adding practice. At the same time, they emphasize the importance of the role of HR professionals, the transformation towards the use of the correct information, and HR analytics in such a way that these contribute to organisational strategy and decision-making.

Table 15 shows the most cited articles existing in the bibliometric database that have also been cited in the references. Continuing among these is the article by [6] as the most cited article with 55 citations, followed by those of [3,37,61] with 50, 38, and 26 citations, respectively.

These are followed by the paper [62] with 25 references. This paper argues that to achieve superior performance and a competitive advantage in companies, HR analytics must be developed as an organisational capacity that is linked to the overall business strategy. This organisational capacity is based on three micro-level categories: individuals, processes, and structure. It further depends on the three dimensions of HR analytics: data quality, analytical competence, and the strategic capacity to act.

The article [63] following this has 23 citations. This study states that HR having an increasing focus on metrics and analytics can help HR functions to take up a larger participatory role in corporate decision-making and strategy creation. Finally, [59] authored the seventh article with 74 citations.

#### 4.2.4. Reference Publication Year Spectroscopy (RPYS)

With this method, a chronological profile of a set of articles is created, highlighting the years with the most significant publications [61] to identify the chronological origins of a discipline. In the time period analyzed, there is an alignment of articles with scientific production as can be seen in Figure 8, highlighting the relevance of years 2010 and 2016 such that these can be considered, of interest in future research on HR analytics, years that are related to the publications by [6,37].

On the other hand, in the analysis of the common terms used in the articles shown in Table 16, in addition to the keywords used to carry out the search for this study, terms were found from the data science area such as “Big Data” and “Artificial Intelligence”, thusdemonstrating that a relationship exists between these terms.

Of these terms, “Big Data” predominates from the time that [6] mention the growing interest in big data shown in HR analytics. Also significant in this regard is the proposal made by [64] that a strategic approach to HR is carried out through the analysis of big data to improve company performance.

Similarly, for the term “Artificial Intelligence”, the paper by [65] reveals that most of the proposed HR analytics models have used artificial intelligence algorithms and methods, demonstrating the rapid development of and the increased interest in applying this technology to the field of HR.

Figure 9 shows the distribution of HR analytics-related themes using the main terms on a map of keywords in the form of a treemap. This represents the most relevant keywords according to the inclusion parameters used in the databases. These are “HR analytics”, “People analytics”, “Workforce analytics”, and “Human Resource analytics”, at 19%, 12%, 6%, and 3% of the total occurrence, respectively.

Additionally, the words “Human Resource Management” and “Analytics” are notable with 6% and 5% of the total occurrence. Similarly, the words “Big Data” and “Artificial Intelligence” are notable with 6% and 4% of the total occurrence respectively. On the other hand, the keyword “Algorithms”, with an occurrence of 1%, shows the lowest prevalence.

Another trend that needs to be analyzed is the behaviour of keywords over time, shown in Figure 10. It can be observed that in the timelines for each keyword, the term “HR analytics” is above the term “People analytics”, although the curve of this latter term tends towards logistic growth in the period analyzed.

In the same way, the word “Big Data” stands out with regard to the terms from the data science area. This should also be noted since the term shows a trend towards greater growth in HR analytics than the other themes do. In contrast, the words “Human Resource analytics” and “Analytics” are notable in showing a decrease, indicating their use in HR analytics articles has lessened.

### 4.3. Analysis of Knowledge Structures

According to [33], three types of general research questions can be answered using bibliometric analysis for scientific mapping to reveal the following:The conceptual structure, to examine the research front for a theme or field of research.The intellectual structure, to identify the knowledge base of a theme or field of research.The social network structure, to discover the production of a particular scientific community.

#### 4.3.1. Conceptual Structure

As shown in Figure 11, conceptual structure is analysed by means of a co-occurrence network using the Louvain clustering algorithm [66,67]. In this, a series of themes related to the main nodes of “HR Analytics” and “People Analytics” are identified. Within these themes, the terms “Big Data” and “Artificial Intelligence” prevail for the “HR Analytics” node, which also coincides with the relationship between the analyses made of the main keywords.

In correspondence to the previous findings, the different themes of a given domain are observed in the thematic map shown in Figure 12. Here, centrality represents the degree of relevance of a field of research, and density represents the degree of development of a theme.

Notable among the terms in the niche topics quadrant are the terms “neural-networks person-organization fit” and “commerce employment”.

In the terms of the motor quadrant, in addition to the main HR analytics themes, there are terms “productivity dynamics”, “diffusion consequences”, “job-satisfaction system”, “future meta-analysis”, “employee turnover”-“human-resources practices”, and “performance-management”.

The emerging or declining themes quadrant contains only the term “Intelligence and Personality”, “job-performance and leadership”, “employees”, and “human resources neural network”.

Finally, in the basic theme’s quadrant, the main themes of “models human”, “privacy issues”, and “work employee perceptions” appear in this order and degree of density.

The thematic evolution of the theme in the period studied is shown in Figure 13. The order of magnitude of the various information flows of quantitative data related to the main themes and the indexing of the content over time are shown via the redundant visualisation of the relationships. This reveals that after 2019 the term “Performance”, “Model”, “employee turnover”, and “future” are united with the term “HR analytics”. Additionally, it shows that the term “HR analytics” mostly became consolidated in its usage between 2020 to 2022.

The Confirmatory Factor Analysis (CFA) approach [68] was used along with the method of Multiple Correspondence Analysis (MCA) [69] to determine the dimensions of this study. Figure 14 shows the two dimensions of HR analytics resulting from this analysis.

The first dimension (27.16%) seems to indicate the level of analysis of the research studies at the HR Analytics level. On the left-hand side of Figure 14 we can see terms focused on the employee and his or her conditions: “employ turnover”, “job satisfaction”, “human resource practices”, “human”, “employment”, or “workplace”. On the right-hand side of this dimension are more generalist terms such as “science”, “innovation”, “management”, “dynamics”, “performance”, “acceptance”. or “human resource analytics”.

On the other hand, the second dimension (14.03%) represents the level of concrete implementation or specialization of the published research studies. At the bottom, there are words such as “employee”, “human resource”, “employment”, or “business analytics”. On the top of this dimension, words like “behavioral”, “adoption”, “strategic”, “impact”, or “firm” shows the level of specialty of this research works.

The dimensional separation shown in Figure 15, using a thematic dendrogram, is consistent with the dimensions that have been identified according to [70,71]. The first branch is related to the main HR analytics terms, which in the association have a height of two, while the following sub-branches have a similar height, thus showing that regardless of the theme, the same domain is being discussed. The other branch of “firm performance” and “information system” has a height of approximately 0.5 and a greater distance between terms, thus confirming the separation of dimensions.

Factor analysis identifies the most cited articles as well as those that make the greatest contribution to each cluster. Figure 16 shows the most cited documents, with the number of links between articles for each theme and for each cluster differentiated by colour. The influence of [3,6,58] in the HR analytics cluster is very significant. However, for the “Analytics” cluster, the opposite happens with very few publications.

#### 4.3.2. Intellectual Structure

The intellectual structure is analysed through a co-citation network [72] and a historiographic map [73]. In the analysis of the co-citation network, the citations of two documents are identified when these are cited by a third document. This is represented graphically as a series of citation occurrences that show a center of gravity as can be seen in the main publications of this study in Figure 17. The centers of gravity of interest for HR analytics are [3,6]; while for “Analytics” they are [59,60]. These are the most influential and co-cited authors in the time period analyzed.

Ref. [59] relates a human capital management (HCM) system to productivity improvement and discuss the advantages and form of implementing an organisational incentive system. On the other hand, present a practical study from which to draw important lessons that show that HR analytics is not a fad in organisational management. The research paper [3] is one of the first contributions as reviews in HR analytics, as it uses an integrative synthesis of published peer-reviewed literature on Human Resource analytics. Ref. [6] highlights the role of Big Data in HR and questions the indispensability of HR Analytics in the strategic management of an organisation. The authors point out that the transformative nature of current HR Analytics practices depends largely on managers and HR professionals being fully aware of its advantages and disadvantages.

Analysis of the historiograph map identifies the research routes and the main authors at different times, as can be seen in Figure 18. In the case of HR analytics, this consolidates into a route with the main authors being [59], followed by [6,60]. However, it is important to note that HR analytics co-citation relationships in recent years have shorter time periods with ranges of around 1 to 3 years with respect to the first years, representing a good sign with respect to the growth and dynamics of this scientific field.

#### 4.3.3. Social Structure

Social structure is analyzed through an examination of the network of co-authors [74] and a map of collaboration between countries. Figure 19 shows the collaboration network, representing the analysis undertaken by the network of co-authors, identifying the authors’ relationships in the field of HR analytics. In this respect, two clusters stand out: the first association of authors to mention is that of Escolar-Jimenez C., Gustilo R., and Matsuzaki K.; this is followed by the association between Singer G., Avrahami D., Pessach D., Chalutz B., and Ben-Gal H. These represent the authors and clusters that collaborated the most in the period analyzed.

With respect to the map of collaboration between countries, Figure 20 shows the relationship lines representing the authors and their countries on the world map for the field of HR analytics. It can be noted that relationships of co-authors between countries in HR analytics happen to a greater extent between the continents of America and Asia. Specifically, a higher frequency of these relationships is identified between authors who collaborate from the countries of the USA and China. Moreover, these countries are followed by other European countries, which feature collaborative co-author relationships between Germany and Spain.

### 4.4. Notable Themes in HR Analytics

Following [75] with respect to bibliometric analysis and a review of the literature, some notable topics for HR analytics were identified from a previous series of papers that carried out systematic reviews of the literature (SLRs) in HR analytics. These investigations are summarized in Table 17.

## 5. Discussion

The results showed that since 2017, scientific production of HR analytics papers has sustained a notable increase, as can be seen in Figure 2. This is possibly due to progress in knowledge in the field as well as awareness of the need to take advantage of technology to generate value using HR information in a way that can influence strategy and managerial decision-making to contribute to improving organisational performance.

The bibliometric analysis of HR analytics conducted expands information on research into this scientific field in combining the Scopus and Web of Science (WoS) databases. This paper analyses a database of 218 articles, whereas similar prior works have analyzed a database of 125 articles [22].

**RQ1.** 
*What are the main themes related to HR analytics?*


It is notable that scientific production in recent years has increased with respect to the first years of the time periodanalyzed. This emerging field of study was also seen to engage in interactions with terms other than those of the main HR analytics themes that were used for this work. Thus, science terms such as “Big Data” and “Artificial Intelligence” are being employed together with the term “Machine Learning” for applications in HR analytics by researchers.

**RQ2.** 
*What are the main scientific journals, authors, and research articles in HR analytics?*


The core sources for HR analytics, shown in the shaded area of Figure 4, are identified by the impact factor of the journals. For HR analytics, the two main scientific journals with the highest impact factor are the journal Human Resource Management, with an h-index of eight, which began publishing on the topic in 2018, and the Journal of Organizational Effectiveness: People and Performance, with an h-index of sevenAND which began publishing on the topic in 2017.

Among the two most cited journals for the topic of HR analytics are the journal Human Resource Management and the International Journal of Human Resource Management, with a total of 212 and 188 citations, respectively. In the growth in journal publications on HR analytics shown in Figure 5, the journal Personnel Review is notable in showing exponential growth. This growth commenced in 2019 and remains on the rise even in the first months of 2022.

In the scientific production of HR analytics authors over the time period studied as seen in Figure 6, the authors with the most publications in HR analytics articles are Escolar-Jimenez C., Gustilo G., and Matsuzaki K. These authors have published fivearticles, with the coincidence that for all threeauthors, the article with the highest number of citations is “A Neural-Fuzzy Network Approach to Employee Performance Evaluation”, published in 2019 with ten citations. This article identifies the use of artificial intelligence techniques in contrast to the subjectivity of the traditional system, which suggests new ways to expand the lines of research applied in HR analytics. 86.3% of the authors, that is, 398 of these, have a single publication in HR analytics.

The previously mentioned authors had the highest impact factor among HR analytics authors, with an h-index of four. Also worth mentioning is the author Boudreau J. with an h-index of three. This author stands out among the HR analytics publications for the co-authorship of the article “An evidence-based review of HR Analytics”.

The most cited articles in HR analytics shown in Table 14 are, in the first place, the article by [6], titled “HR and analytics: Why HR is set to fail the big data challenge”, with 147 citations and an average citation rate per year of 21 times. This is followed by the article by [57] titled, “Are we there yet?: What’s next for HR?”, with 122 citations and an average rate of citations per year of 15.25 times. These amounts could be considered small compared to other topics. However, for this topic, it is very relevant to know the article by [6]), as it is also quite influential because it is the most cited reference in Table 14.

**RQ3.** 
*How has the concept of HR analytics developed in recent years?*


The co-occurrence network shown in Figure 11 is used to analyze the conceptual structure, demonstrating the prevalence of the main node of “HR Analytics” with the terms of “HR Analytics”, “Big Data”, and “Artificial Intelligence”. Respectively, these have 19%, 6% and 4% of the total occurrence of the keywords in the form of a treemap shown in Figure 9.

The Confirmatory Factor Analysis shown in Figure 14 identifies the main dimension of this study by the terms “HR Analytics”, “People Analytics”, and “Workforce Analytics”; these together with the terms “Big Data”, “Artificial Intelligence”, and “Human Resource Management” maintain an association that represents 82.64% of the cases in this dimension.

The intellectualstructure is analyzed using the co-citation network shown in Figure 17 and the historiographic map shown in Figure 18. These identify the important centres of gravity for HR analytics to be [3,6]. In addition, HR analytics co-citation relationships in recent years have shorter time periods with respect to earlier years, now featuring ranges of around 1 to 3 years. This is a good sign of the growth and dynamics of this scientific field.

Analysis of social structure in the field of HR analytics is made through the network of co-authors shown in Figure 19 and the map of collaboration between countries shown in Figure 20. These highlight the cluster with the strongest association as being that of the authors Escolar-Jimenez C., Gustillo R., and Matsuzaki K. Relationships in HR analytics between co-authors in different countries occur to a greater extent between authors collaborating in the countries of the USA and China.

Scientific production in HR analytics by country is led by the USA with 43 articles. This is also the country showing the highest rate of collaboration with an MCP of four. It is followed by India with twenty-fivearticles and an MCP of threein terms of its collaboration rate.

In the same way, the USA maintains the highest number of article citations per country at 933 citations, representing an average of 21.7% of all citations. It is again followed by India with 223 citations, representing an average of 8.92% of all citations.

**RQ4.** 
*What is the focus and vision of future research in HR analytics?*


The summary of notable HR analytics themes revealed by the systematic review of the literature (SLR) as shown in Table 16 seeks to give rise to opportunities to promote the closing of gaps in HR analytics. These are proposed to promote progress in the development of research on this subject and to capture recommendations for topics of interest for future exploration.

The authors of [17] propose the balance of interest approach to explore the theoretical perspective at the individual, team, and organisational level, in order to further extend HR analytics research, which has necessarily concentrated on the application of HR analytics, reinforcing the premise that empirical work iscarried out to demonstrate the theoretical relationship, the antecedents of HR analytics and the general performance of the organisation.

Works such as the benchmark paper by [49] have explored such topics, indicating that the adoption of HR analytics improves through the incorporation of return on investment (ROI) analysis or an ROI-based framework. This paper further emphasizesthe context in which HR analytics is being adopted and implemented, both in practice and in theory.

The frameworks that describe the adoption of innovation according to [3] can serve as a basis for understanding the current situation regarding the adoption of HR analytics and its probable future. And likewise, for example, so do the theoretical frameworks that are related to strategic management and organisationalbehaviour.

Furthermore, to understand and contextualise HR analytics as an innovation in HRM, [76] have used the theory of planned behaviour, the diffusion model of innovation and the technology-organisation-environment framework to subsequently provide a framework for the adoption of HR analytics that identifies five factors influencing this in any organisation, these being technological, organisational, environmental, data governance, and individual factors.

However, the application of HR analytics depends on driving a proactive HR research and analytics agenda in terms of enabling strategic HR decisions. Therefore, it is necessary for an applied researcher with a background in the social, behavioral, and organisational sciences to accurately and ethically interpret the insights derived from HR analytics in the context of individual, group, and organisational behavior [78,79].

Finally, the use of Artificial Intelligence (AI) learning algorithms, allowed [21] to identify the dangers related to the application of HR analytics. In summary, therefore, we can say that HR analytics is a discipline that uses data and analytical tools to make informed decisions about employee management and organisational performance. Some of the main practical applications of HR analytics are Employee selection and recruitment, helping identify the most suitable candidates for a job using psychometric tests, resume analysis. and structured interviews; Performance evaluation supporting measure employee performance, identify areas for improvement and set clear objectives for skills development and promotion; Talent retention, identifying employees who are most at risk of leaving the organisation and develop strategies to retain them, such as career development programmes and additional benefits; Workforce planning: an organisation forecast future staffing needs, identify skills gaps, and develop plans to address them; Training programme design: planning the skills that employees need and developing training programmes that are effective in meeting those needs.

## 6. Conclusions

This bibliometric analysis of the scientific literature on HR analytics has made it possible to affirm that the area continues to emerge and to incorporate new terms of interest from the area of data science. At the same time, it is very adaptive due to the need to access personal information through HR information systems and databases to be used in a utilitarian and ethical way by companies for the benefit of the employees themselves as well as organisations.

It, therefore, provides the focus and current state regarding the terms that are most recently used in HR analytics with respect to the search criteria applied to carry out the research into the state-of-the-art of this discipline. Likewise, it emphasizes the value of the current state of scientific production with articles published up to 2022, demonstrating that the field remains dynamic, emerging and trending in accordance with [3,6,61].

For organisations, the digital transformation of HR and traditional HR practices with approaches employing technological innovations has made promotion of the use of HR information into a current pressing need to improve the strategies and the performance of organisations themselves, as well as of the people forming part of them. The paper by [80] seeks to contribute to HR digitisation literature through the adoption of HR analytics.

The benefits for people and organisations can be seen in the usefulness of opting for better performance in the so-called Industry 4.0 (or fourth industrial revolution) by using the information available for decision-making with the application of HR analytics to achieve strategy and business objectives. In addition, HR analytics is postulated as an innovation in HRM, which can accelerate organisational changes, motivating business digitisation in a way always linked to people, forming an intangible value within the very identity and culture of companies.

The incorporation of future research that analyses the adoption and implementation of HR analytics empirically with quantitative studies made using adoption frameworks could further expand knowledge on the subject over and above successful business cases, which allow the analysis of the subject taking into accountorganisational performance itself and its relationship with other variables of interest. This could be either to learn the level of innovation employed or the increase in sales of companies achieved through improving the performance of their employees. Such applications could quantitatively establish these new strategic HR practices for industries at the managerial level and for decision-making based on data, with the novelty of being modern and technological.

Thus, an empirical examination of the adoption of HR analytics could highlight or help expand that understanding, as has been done in similar technology adoption analysis studies [81].

Within the practical limitations of this research into HR analytics is the acquisition, use, and knowledge of the technology itself, given that other areas of companies remain in processes of digital transformation. Without this being an end in itself, customers and employees themselves push organisations into accelerated updating processes to remain in the market, as a strategy to maintain their own survival [82]. Another limitation has been to deal with a lot of scattered information limited to specific issues, such as HR Analytics, which does not favour a general overview, although it does favour a description of the situation of scientific research in this specific field.

The field of research into HR Analytics remains of great interest;however, the adaptability of other topics according to their own dynamics sees the body of researchers also evolve in like fashion over time. Similarly, the depth of the subject matter can lead to other turns of research and interests due to aspects related to the main topic, leading this to instead focus on more specific themes, so expanding the subject with terms from the data science area such as “Big Data”, “Artificial Intelligence” [83], and “Machine Learning” that are currently being taken up in the application of HR analytics.

In the future, more research will be required in the field of HR analytics due to an increasingly technological world that at an organisational level could benefit further from this in its own performance, whether these are large companies or small or medium-sized ones. The breadth of the topic of HR analytics should thusbe investigated more thoroughly in all its aspects and variations, especially with regard to its applications in different areas by researchers and data scientists, as well as from within or as part of corporations themselves. One of the fields within HR Analytics will be the study of telework performance [84].

The limitations of this bibliometric study are the collection of bibliographic metadata in the Scopus and Web of Science (WoS) databases. This study is limited to these databases.

In short, this research could also be of great interest to academics and professionals who seek to discover the-state-of-the-art of this topic, as well as to expand contributions to knowledge in this scientific field. In this article, bibliometric analysis was employed to identify the main authors contributing their knowledge to the field of HR analytics.

## Figures and Tables

**Figure 1 behavsci-13-00244-f001:**
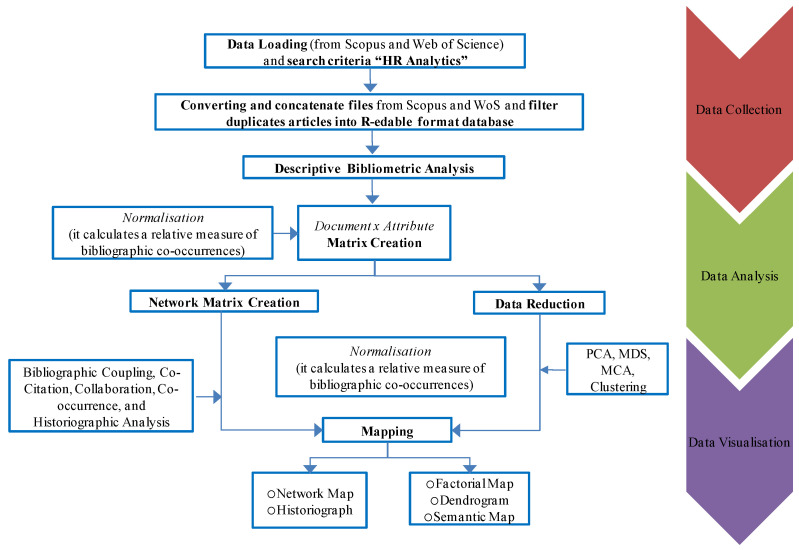
Bibliometrix and the Recommended Science Mapping Workflow.

**Figure 2 behavsci-13-00244-f002:**
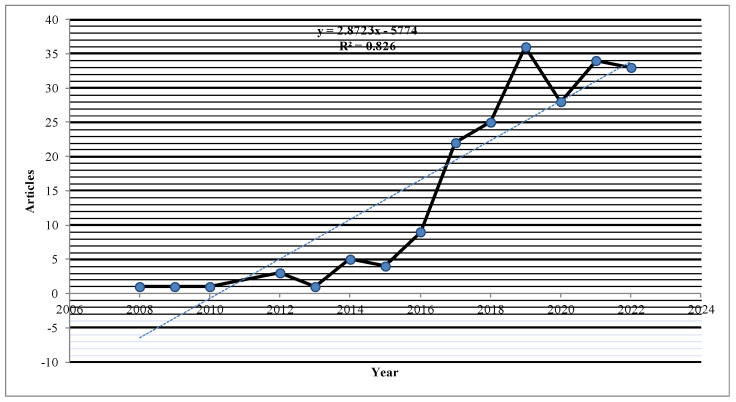
Annual Scientific Production.

**Figure 3 behavsci-13-00244-f003:**
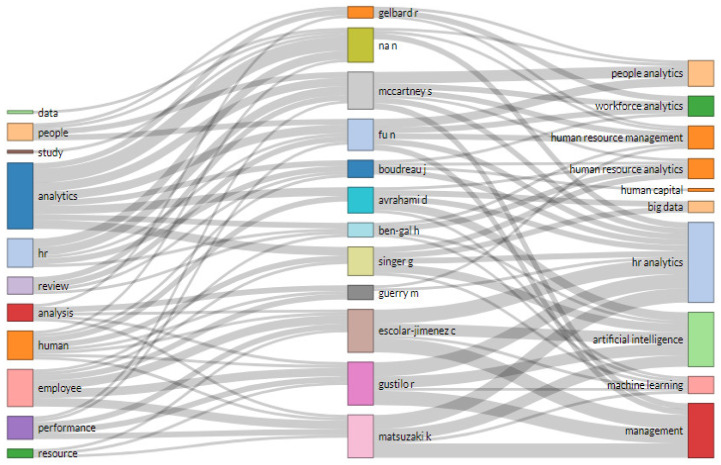
Thematic Development.

**Figure 4 behavsci-13-00244-f004:**
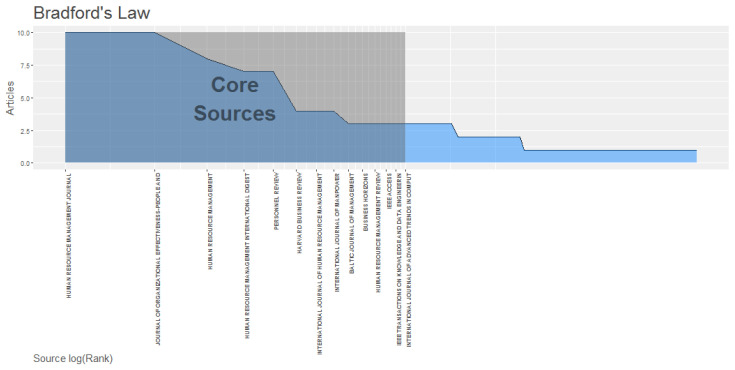
Core Sources.

**Figure 5 behavsci-13-00244-f005:**
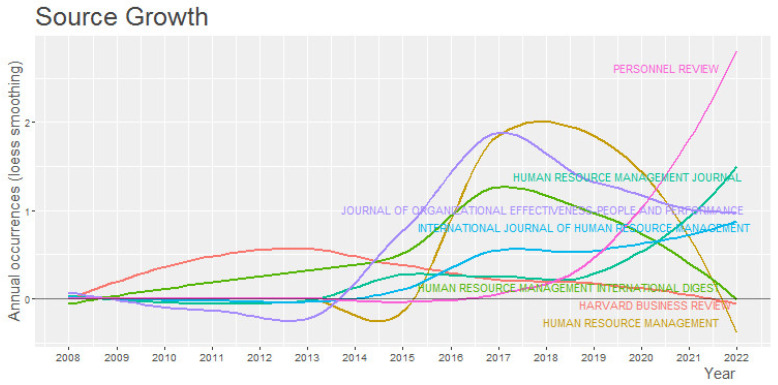
Growth of Journal Publications.

**Figure 6 behavsci-13-00244-f006:**
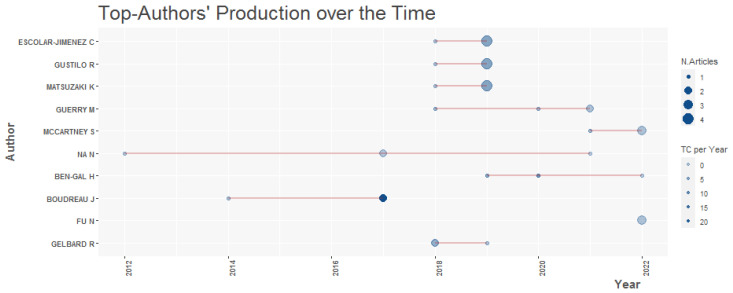
Scientific Production of Authors over Time. Other authors grouped under the acronym “NA N”.

**Figure 7 behavsci-13-00244-f007:**
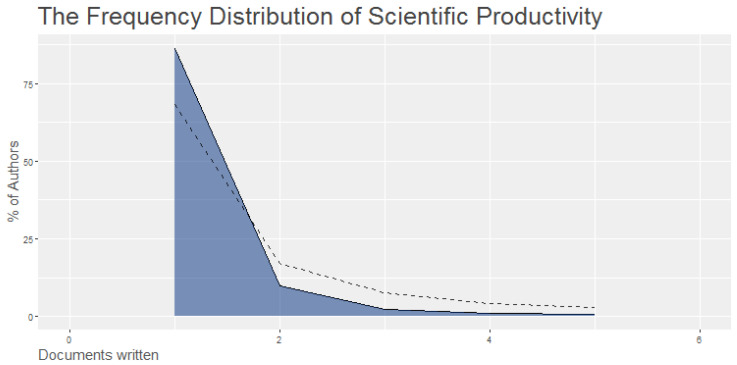
Scientific Production of Publications.

**Figure 8 behavsci-13-00244-f008:**
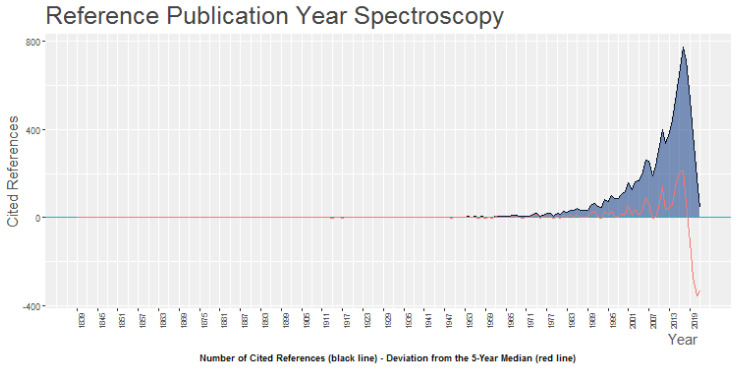
Reference Publication Year Spectroscopy (RPYS).

**Figure 9 behavsci-13-00244-f009:**
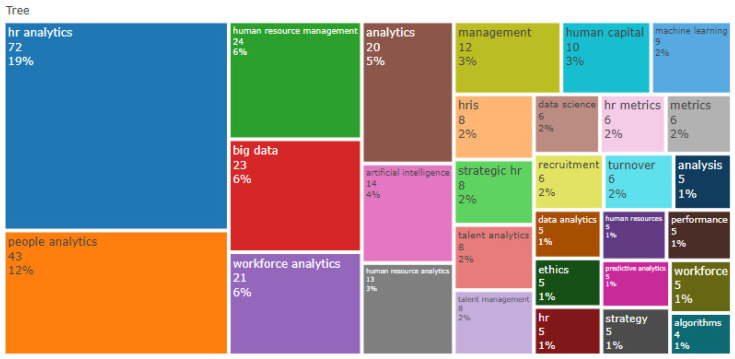
Word TreeMap.

**Figure 10 behavsci-13-00244-f010:**
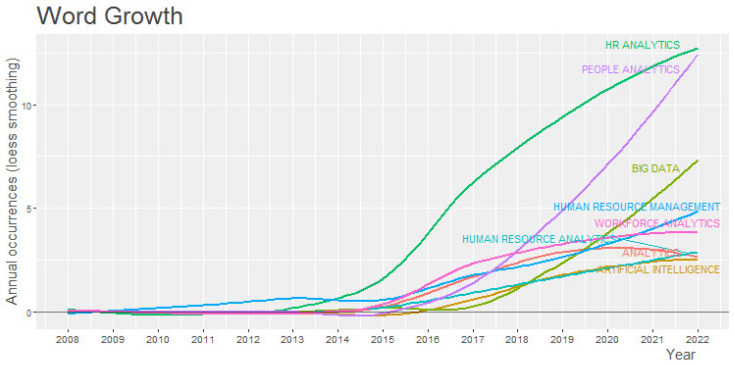
Keyword Growth over Time.

**Figure 11 behavsci-13-00244-f011:**
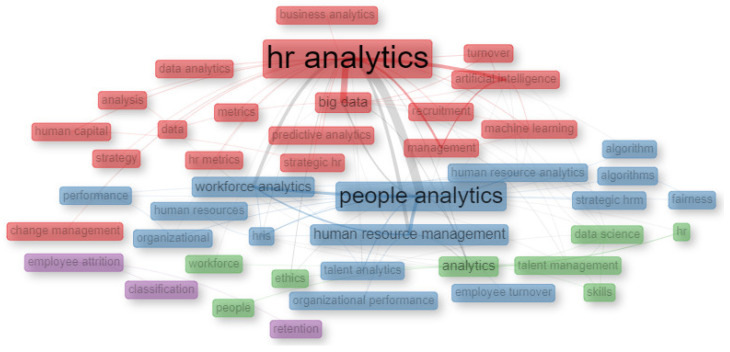
Co-Occurrence Network.

**Figure 12 behavsci-13-00244-f012:**
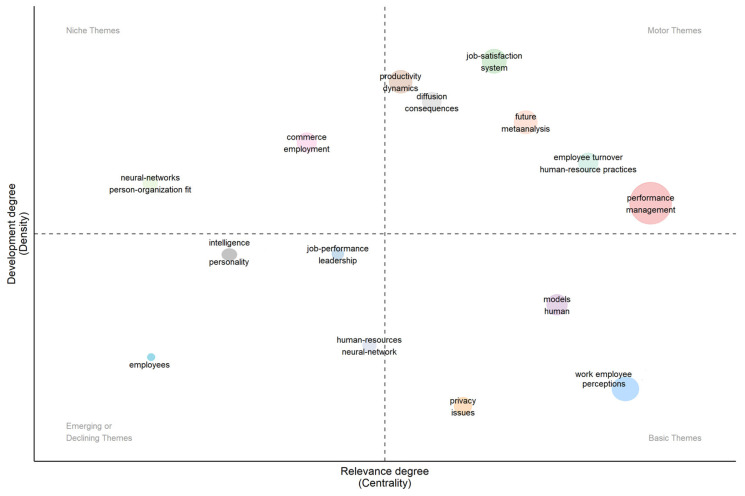
Thematic Map.

**Figure 13 behavsci-13-00244-f013:**
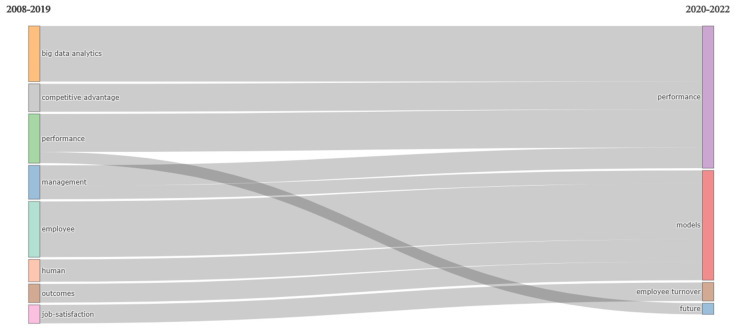
Thematic Evolution.

**Figure 14 behavsci-13-00244-f014:**
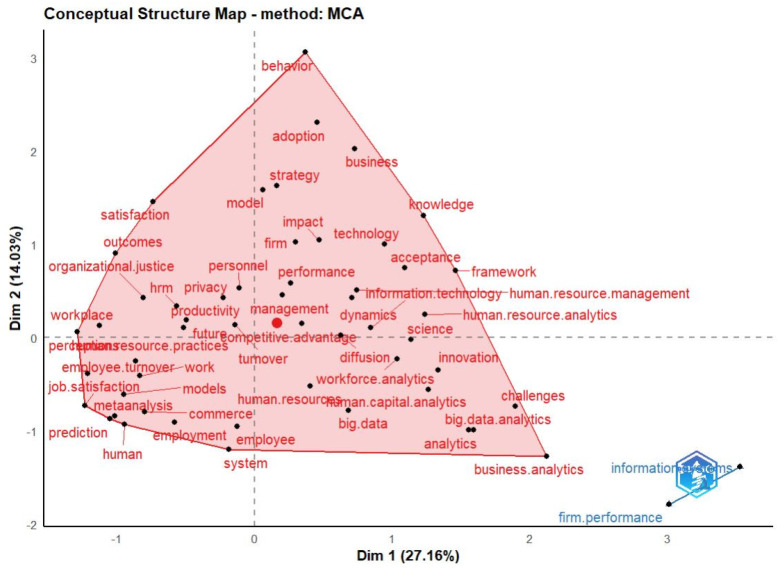
Factorial Analysis (MCA).

**Figure 15 behavsci-13-00244-f015:**
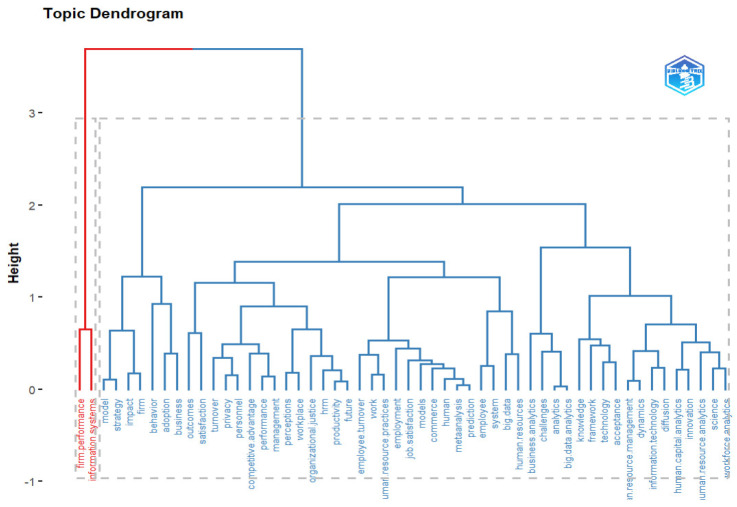
Thematic Dendrogram.

**Figure 16 behavsci-13-00244-f016:**
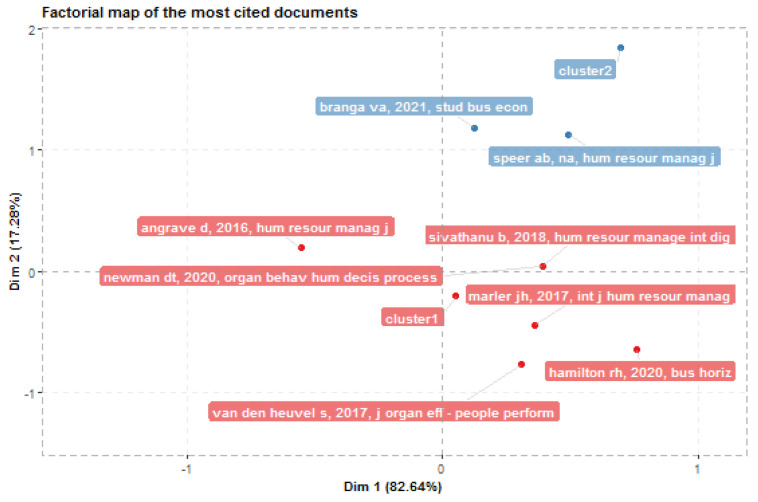
Most Cited Articles.

**Figure 17 behavsci-13-00244-f017:**
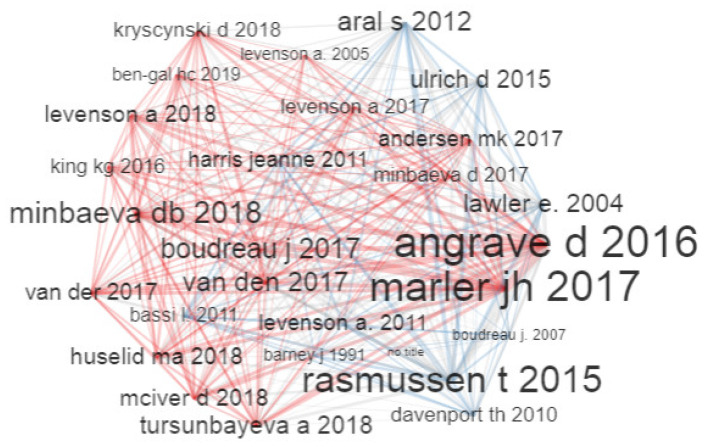
Co-Citation Network.

**Figure 18 behavsci-13-00244-f018:**
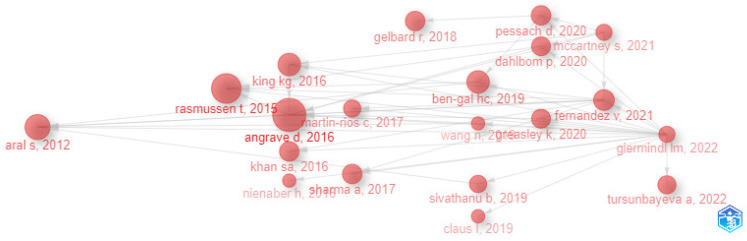
Historiograph Map.

**Figure 19 behavsci-13-00244-f019:**
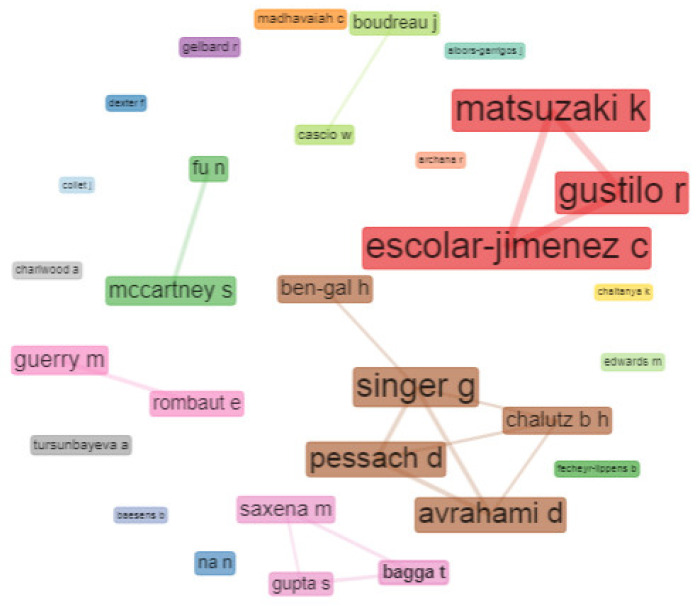
Collaboration Network.

**Figure 20 behavsci-13-00244-f020:**
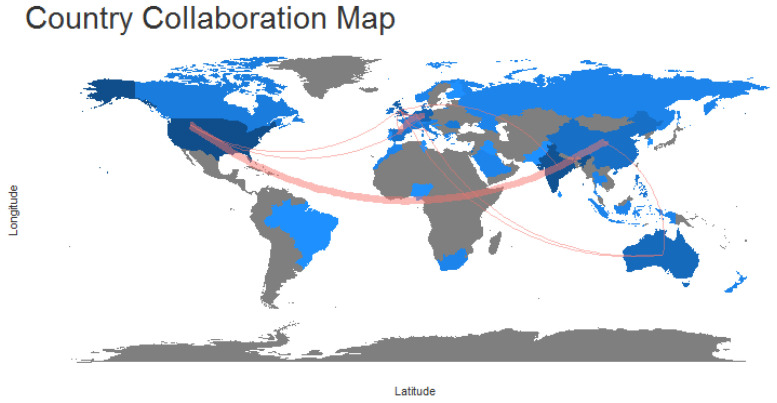
Collaboration World Map.

**Table 1 behavsci-13-00244-t001:** Research Questions.

	Research Question	Objective	Motivation
RQ1	What are the main themes related to HR analytics?	To present the main themes addressed by researchers	To discover the core themes of HR analytics
RQ2	What are the main scientific journals, authors, and research articles in HR analytics?	To identify the most relevant sources, authors, and articles	To contribute to improving the understanding of HR analytics
RQ3	How has the area of HR analytics developed in recent years?	To analyze the evolution of the conceptual, intellectual, and social structure	To expand the understanding of HR analytics
RQ4	What is the focus and vision of future research in HR analytics?	To provide guidance as to possible notable research themes as well as those of future interest	To provide possible future HR analytics themes

**Table 2 behavsci-13-00244-t002:** Previous Literature Overviews.

Authors	Type/Period	Data Sources	Context	Screened Works/Primary Studies	Methodology Based
[3]	SLR 2000–2016	Academic Search Complete, Business Source Complete, and Scopus	This evidence-based review uses an integrative synthesis of published peer-reviewed literature on Human Resource analytics (HR Analytics).	14/60	[10]
[11]	SLR unspecified-2021	Web of Science, Scopus, and PsycINFO)	This work analyzes the current state of HRA and proposes a framework for the development of HRA as a sustainable practice.	34/423	[12]
[13]	SLR unspecified-2019	Scopus	A systematization of research topics and directions for future research about Human Resources analytics. This work uses a systematic literature review process and deconstructs the concept of HRA as presented in the literature, which identifies 106 key research topics associated with three major areas, i.e., enablers of HR analytics (technological and organizational), applications (descriptive and diagnostic/prescriptive), and value (employee value and organizational value).	68/301	[14,15]
[16]	SLR unspecified-August 2021	Scopus	A literature review identifies and synthesizes existing literature on people analytics and its conceptualised efficacy. The objective is to explore and understand the efficacy of people analytics to enable the HR function to become a strategic partner.	90/671	[12]
[17]	SLR 2011–2021	ABI Inform, Business Source complete, Emerald, Scopus, and Wiley Online Library	This study conducts a systematic literature review of peer-reviewed articles focused on people analytics in the Association of Business School (ABS) Index aims to investigate the current reality of people analytics and uncover the debates and challenges that are emerging as a result of its adoption.	46/2725	[3,4,13]
[18]	Bibliom.2013–2021	Scopus	Bibliometric Review of People Analytics.VOSviewer constructs and visualizes bibliometric networks, including articles, conference papers, book chapters, editorials, notes, and reviews.	127/129	[3]

**Table 3 behavsci-13-00244-t003:** Search Criteria in the Databases.

Database	Search Criteria	Total
Web of Science	(TS = (“People analytics” or “HR analytics” or “Human Resource analytics” or “Workforce analytics” or “Talent analytics” or “Employee analytics” or “Human Capital analytics”)) AND (DT = (“ARTICLE” OR “EARLY ACCESS” OR “REVIEW”) AND LA = (“ENGLISH”))	138
Scopus	TITLE-ABS-KEY (“People analytics” OR “HR analytics” OR “Human Resource analytics” OR “Workforce analytics” OR “Talent analytics” OR “Employee analytics” OR “Human Capital analytics”) AND (LIMIT-TO (DOCTYPE, “ar”) OR LIMIT-TO (DOCTYPE, “re”)) AND (LIMIT-TO (LANGUAGE, “English”))	193

**Table 4 behavsci-13-00244-t004:** Main Information.

Description	Results
Time period	2008:2022
Sources (Journals, Books, etc.)	134
Articles	218
Average publications per year	3
Average citations per article	10.44
Average citations per article per year	2.437
References	9390
Document Type	
Articles	183
Early access	15
Review	20
Content of the Documents	
Keywords Plus (ID)	473
Author’s keywords (DE)	652
Authors	
Authors	461
Author appearances	551
Single authors	41
Multiple authors	420
Collaboration	
Single author	45
Documents per Author	0.473
Authors per document	2.11
Co-Authors per article	2.53
Collaboration Index	2.43

**Table 5 behavsci-13-00244-t005:** Most Relevant Scientific Sources.

Sources	Articles
Human Resource Management Journal	10
Journal of Organizational Effectiveness: People and Performance	10
Human Resource Management	8
Human Resource Management International Digest	7
Personnel Review	7
Harvard Business Review	4
International Journal of Human Resource Management	4
International Journal of Manpower	4

**Table 6 behavsci-13-00244-t006:** Most Cited Sources.

Sources	Articles
Human Resource Management	212
International Journal of Human Resource Management	188
Academy of Management Journal	154
Harvard Business Review	144
Journal of Applied Psychology	138
Human Resource Management Review	129
Journal of Organizational Effectiveness: People and Performance	123
Journal of Management	122
Human Resource Management Journal	120
Academy of Management Review	109

**Table 7 behavsci-13-00244-t007:** Journal Impact.

Source	h_index	g_index	m_index	TC	PY_start
Human Resource Management Journal	3	10		171	2016
Journal of Organizational Effectiveness: People and Performance	7	10		144	2017
Human Resource Management	8	8	1.6	168	2018
Human Resource Management International Digest	3	7	0.27273	142	2012
Personnel Review	4	6		44	2019
Harvard Business Review	4	4	0.30769	193	2010
International Journal of Human Resource Management	3	4	0.5	134	2017
International Journal of Manpower	2	3		14	2020

Note: TC: Times Cited, PY_start: Publication start year.

**Table 8 behavsci-13-00244-t008:** Relevant Authors.

Authors	Articles	Area
Caryl Charlene Escolar-Jimenez	5	Computer Science
Reggie C. Gustilo	5	Computational Intelligence
KichieMatsuzaki	5	Industrial and Management Systems Engineering
Marie-Anne Guerry	4	Business Technology and Operations
Steven McCartney	4	Management and Organisational Behaviour
Others (*)	4	Others
Gonen Singer	4	Industrial Engineering and Data Science
Dan Avrahami	3	Data Science
Hila Chalutz Ben-Gal	3	Industrial Engineering and Management
John Boudreau	3	Human Resources

(*) Other authors grouped under the acronym “NA N”.

**Table 9 behavsci-13-00244-t009:** Distribution of Scientific Production According to Lotka’s law.

	Articles Written	No. of Authors	Proportion	No. of Publications
	1	398	86.3%	398
	2	46	10.0%	92
	3	10	2.2%	30
	4	4	0.9%	16
	5	3	0.7%	15
	n	n	1/n^2^	N
Total		n	100%	N

**Table 10 behavsci-13-00244-t010:** Impact Factor of the Authors.

Author	h_index	g_index	m_index	TC	NP	PY_start
Escolar-Jimenez C.	4	5	0.8	31	5	2018
Gustilo R.	4	5	0.8	31	5	2018
Matsuzaki K.	4	5	0.8	31	5	2018
Guerry M.	2	4	0.4	24	4	2018
McCartney S.	1	2	0.5	6	4	2021
Others	1	1	0.091	3	4	2012
Singer G.	1	4		28	4	2020
Avrahami D.	1	3		28	3	2020
Ben-Gal H.	2	3	0.5	50	3	2019
Boudreau J.	3	3	0.333	150	3	2014

Note: TC: Times Cited, NP: Number of publications, PY_start: Publication start year.

**Table 11 behavsci-13-00244-t011:** Affiliations of the Authors.

Affiliations	Articles
Bar-Ilan University	10
Tilburg University	8
University of Southern California	8
Copenhagen Business School	6
De La Salle University	5
Katholieke Universiteit Leuven	5
Lucian Blaga University of Sibiu	5

**Table 12 behavsci-13-00244-t012:** Scientific Production by Country.

Country	Articles	Freq	SCP	MCP	MCP_Ratio
USA	43	0.25444	39	4	0.093
India	25	0.14793	22	3	0.12
United Kingdom	11	0.06509	9	2	0.182
Germany	8	0.04734	6	2	0.25
Israel	8	0.04734	6	2	0.25
Netherlands	8	0.04734	6	2	0.25
Australia	7	0.04142	4	3	0.429
Belgium	5	0.02959	2	3	0.6
Ireland	5	0.02959	5	0	0
Spain	5	0.02959	3	2	0.4

Note: Freq: Frequency; SCP: Single country publications; MCP: Multiple country publications; MCP Ratio: Multiple country publications ratio.

**Table 13 behavsci-13-00244-t013:** Average Number of Article Citations per Country.

Country	TC	AAC
USA	933	21.70
India	223	8.92
United Kingdom	204	18.55
Netherlands	164	20.50
Denmark	84	21.00
Israel	77	9.62
Australia	68	9.71
Belgium	63	12.60
Italy	50	16.67
Spain	45	9.00

Note: TC: Times Cited, AAC: Average Article Citations.

**Table 14 behavsci-13-00244-t014:** Most Cited Articles.

Paper	Title	DOI	TC	TCY
Angrave et al. (2016) [6]	HR and analytics: Why HR is set to fail the big data challenge	10.1111/1748-8583.12090	147	21
Ulrich & Dulebohn (2015) [57]	Are we there yet? What’s next for HR?	10.1016/j.hrmr.2015.01.004	122	15.25
Sivathanu & Pillai (2018) [58]	Smart HR 4.0—Hindustry 4.0 is disrupting HR	10.1108/HRMID-04-2018-0059	121	24.2
Davenport et al. (2010) [37]	Competing on Talent Analytics	NA	117	9
Marler & Boudreau (2017) [3]	An evidence-based review of HR analytics	10.1080/09585192.2016.1244699	113	18.833
Aral et al. (2012) [59]	Three-Way Complementarities: Performance Pay, Human Resource Analytics, and Information Technology	10.1287/mnsc.1110.1460	101	9.182
Rasmussen & Ulrich (2015) [60]	Learning from practice: How HR analytics avoids being a management fad	10.1016/j.orgdyn.2015.05.008	74	9.25

Note: TC: Times Cited, TCY: Times Cited per year. NA: not assigned.

**Table 15 behavsci-13-00244-t015:** Most Cited References.

Cited References	Title	DOI	Citations
Angrave et al. (2016) [6]	HR and analytics: Why HR is set to fail the big data challenge	10.1111/1748-8583.12090	55
Marler & Boudreau (2017) [3]	An evidence-based review of HR analytics	10.1080/09585192.2016.1244699	50
Rasmussen & Ulrich (2015) [60]	Learning from practice: How HR analytics avoids being a management fad	10.1016/J.ORGDYN.2015.05.008	38
Davenport et al. (2010) [37]	Competing on Talent Analytics	NA	26
Minbaeva (2018) [62]	Building credible human capital analytics for organisational competitive advantage	10.1002/HRM.21848	25
Lawler et al. (2004) [63]	HR metrics and analytics: Use and Impact	NA	23
Aral et al. (2012) [59]	Three-Way Complementarities: Performance Pay, Human Resource Analytics, and Information Technology	10.1287/MNSC.1110.1460	21

Note: DOI: Digital Object Identifier. NA: not assigned.

**Table 16 behavsci-13-00244-t016:** Keywords.

Words	Occurrences
HR analytics	72
People Analytics	43
Human Resource Management	24
Big Data	23
Workforce Analytics	21
Analytics	20
Artificial Intelligence	14
Human Resource Analytics	13

**Table 17 behavsci-13-00244-t017:** Summary of notable themes in HR analytics revealed by SLRs.

Paper	Title	Period	Overview of the Highlights
[3]	An evidence-based review of HR analytics	2000–2015	Investigation of the adoption and use of HR analytics as well as the moderators and nomological networks for HR analytics
[4]	People analytics: A scoping review of conceptual boundaries and value propositions	2002–2017	Conducts studies of evaluation, implementation cases, and model simulation in HR analytics
[49]	An ROI-based review of HR analytics: Practical implementation tools	2000–2016	Improves and develops empirical and conceptual knowledge on cutting-edge tools for HR analytics
[22]	Human resource analytics: A review and bibliometric analysis	2008–2019	Performs a metadata analysis of HR analytics in WoS and a quantitative analysis that allows SLR analysis
[50]	The ethics of people analytics: Risks, opportunities and recommendations	2006–2019	Develops theoretical ethical guidelines for HR analytics
[76]	Examining the determinants of successful adoption of data analytics in human resource management: A framework for implications	1994–2020	Empirically tests the framework of HR analytics adoption with quantitative and qualitative studies. Studies prior research, moderators, and the results of HR analytics. Identifies the actors influencing the adoption of HR analytics
[21]	The dark sides of people analytics: Reviewing the perils for organisations and employees	Before 2020	Expands analysis of the negative consequences of HR Analytics
[77]	An operational conceptualisation of human resource analytics: Implications for in human resource development	1990–2021	Conducts HR analytics studies from the perspective of employees, as well as on the competencies of HR professionals
[17]	Promise versus reality: A systematic review of the ongoing debates in people analytics	2011–2021	Examines the impact and success of HR analytics analysis at individual, team, and organisational levels through theoretical lenses

## Data Availability

Not applicable.

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
