# Peer review of "Exploring the Evolution of Human Resource Analytics: A Bibliometric Study"

_behavsci, 2023, doi:10.3390/bs13030244_

Round 1

Reviewer 1 Report

Review of “Exploring the evolution of human resource analytics: a
bibliometric study”

Thanks for providing me with the opportunity to read your manuscript which deals with a very timely topic. The overall aim of the manuscript is interesting, but it has a few shortcomings that should be addressed. Below follow some comments and suggestions that I hope could be useful in a future revision.

·         You should clearly state the contribution and added-value of the study since there are some previous bibliometric analyses of HR analytics (that you cite). You state that your data set differs, but are your analyses different in any way?  

·         The literature review section (2) is too short, only about 1 page. Maybe some things can be moved from the introduction to the literature review. The authors should also provide some more background to the concept of HR analytics and discuss and reflect on studies of its use/popularity in the business world. You write a bit about Google's role in popularizing People analytics. See for example [1-3]

·         In the methods section, you could discuss why Biblioshiny is better suited for your purposes than the other software packages available (e.g., VosViewer, Citespace).

·         The authors present the results of many different bibliometric analyses that can be done in Biblioshiny. A danger is that too much is packed into one paper. For example, you attempt to examine the conceptual, intellectual, and social structure of the research field. This is a lot. However, the part about intellectual structure (4.3.2) is short and a bit superficial. Maybe reconsider if some of them can be dropped and instead go deeper into the rest. Or argue more convincingly why they are all needed in this one paper.

·         You should provide some more detailed descriptions of various tables (e.g. Table 4/5).

·         In Figure 12, you should, in my opinion, consider merging synonymous terms such as “hr analytics” and human resource analytics”. This can be easily quite done in the Biblioshiny software by uploading a .txt file. Same goes for Fig 13.

·         There are many short paragraphs (2-3 lines), especially in Section 3. Maybe some of these can be integrated to improve the flow of the paper.

·         Table 4: “Arts”, I assume it is short for “articles” but not clear.

·         You should add some more reflections about the limitations of your study and the use of bibliometric techniques more generally.

Good luck!

 References

1.            Marler, J.H.; Cronemberger, F.; Tao, C. HR Analytics: Here to Stay or Short Lived Management Fashion? In Electronic HRM in the Smart Era, Bondarouk, T., Ruël, H.J.M., Parry, E., Eds.; Emerald Publishing Limited: 2017; pp. 59-85.

2.            Madsen, D.Ø.; Slåtten, K. An Examination of the Current Status and Popularity of HR Analytics. International Journal of Strategic Management 2019, 19, 17-38.

3.            Van den Heuvel, S.; Bondarouk, T. The rise (and fall?) of HR analytics: a study into the future application, value, structure, and system support. Journal of Organizational Effectiveness: People and Performance 2017, 4, 157-178.

Author Response

Thank you for this comment. We appreciate all your comments and efforts to improve this manuscript's quality. You will find the authors' response in the attached file.

Reviewer 2 Report

This is a very interesting topic to explore the evolution of human resource analytics through bibliometric analysis.

1 As a new trend in business practice, HR analytics emphasizes the application of HR-related data to help organizations make decisions. However, the focus on data has been agreed upon in academic research, and the authors need to explain clearly in the introduction the similarities and differences between HR analytics in practice and in academic research.

2 The literature review should not only introduce the operation and procedures of HR analytics, but also focus on the current studies on HR analytics.

3 Is it accurate to define scientific work in HR analytics simply by having HR analytics or "People analytics" in the title or keywords? In fact, a large number of qualitative studies involve HR analytics.

4 I am a little confused whether the authors are focusing on practical articles using HR analytics, or academic articles using HR analytics as a research method?

5 In terms of data, the most prolific authors in HR analytics are mainly from the fields of computer science and data science. Is this due to the fact that organizational behavior and HRM research equally data-focused, but use different names (e.g., empirical studies? Questionnaires?). How are empirical studies and HR analytics defined and differentiated?

6 In the discussion section it is suggested to explore more the application of HR analytics to academic research, especially in the OB and HR fields.

Author Response

(The authors gave the same response as above.)

Round 2

Reviewer 1 Report

I would to thank the authors for providing point-by-point responses to the comments and recommendations. The revised version has been improved. At this point, I recommend some editing and proofreading to improve the flow and readability of the paper. 

Reviewer 2 Report

The authors have mostly responded my concerns, and I think that the manuscript has been improved to be considered for publication in Behavioral Sciences.